# Attitudes of Drivers towards Electric Vehicles in Kuwait

**Andri Ottesen [1],\*, Sumayya Banna [2] and Basil Alzougool [2]**

1   LSE Middle East Centre (MEC)—Sustainability Research and Consultancy (CSRC), Australian University, West Mishref P.O. Box 1411, Kuwait
2   LSE Middle East Centre (MEC), Arab Open University, Ardiya P.O. Box 830, Kuwait
\*   Correspondence: a.ottesen@au.edu.kw

**Abstract:** Although researchers have started to examine the landscape of electric vehicles (EVs) around the world, very little research has examined this phenomenon in Kuwait. In addition, limited research has explored it among drivers. Kuwait constitutes a very promising market for EVs because there is a need to lower GHG emissions and improve the air quality in Kuwait. This study therefore explored the attitudes of conventional car internal combustion engine (ICE) drivers towards EVs in Kuwait, particularly identifying attributes, features, enablers, and barriers of EVs that are considered important by potential consumers in Kuwait. This study utilized a mixed method approach in terms of quantitative data and qualitative data from a sample of 472 drivers to accomplish the main objectives of this study. The study showed that more than half of participants would buy an EV within the next 3 years, and they would buy if several conditions were met. That includes a cheaper purchase price with the assistance of policies controlled by the government along with the availability of suitable infrastructure for EVs relating to charging stations, fast lanes, and free parking spaces. More than 40% of participants would also seriously start thinking about buying an EV if the gas/fuel prices increased by between 50 and 199%. More than 40% of participants thought that EVs are safe in relation to fire and car crashes. Furthermore, approximately half of participants would pay 6–20% more for an EV that is both environmentally friendly and much quicker than gasoline cars. In addition, participants would also prefer EVs over gasoline cars in the future for their environmental, economic, and technological values. More importantly, the study yielded many significant findings, such as the demanded and preferred features of EVs and reflections on the readiness of the Kuwaiti market.

**Keywords:** electric vehicle; sustainability; sustainable mobility; circular economy; consumers; drivers; attitudes; barriers; marketing; Kuwait

## 1. Introduction

The burgeoning issue of sustainable mobility has grown in recent years [1–8]. As our planet faces the consequences of climate change, including the COVID-19 crisis, actions are needed to adapt to a more sustainable environmental development model inclusive of greener and cleaner technologies in each sector [9–13]. Green technologies can be described as those cleaner technologies that are capable of protecting the natural environment by reducing air and noise pollution and conserving energy and resources [14–17]. During the last three decades, the corporate sector, for instance, has already started to adapt to a more sustainable development model, and to integrate sustainability into their business operations by using eco-friendlier solutions, resource-efficient technologies and greener technologies, and achieving optimal levels of circular economy [15]. Therefore, it becomes urgent to implement sustainable mobility in the transportation sector, as it links between land use and transport systems, and to encourage greater efficiency in the transport system by shifting to a more sustainable model [8,15,18–20].

The transportation sector has negatively impacted the environment and health of people, causing many diseases via the fossil fuels that contribute to one-sixth of the worldwide greenhouse gas emissions [17,21]. Therefore, the adoption of electric vehicles (EVs) as

cleaner and greener technologies has been identified to be energy efficient and effective in combating greenhouse gas emission (GHG) and noise, as a way to shift towards sustainable mobility [18–21]. The transportation market is mainly driven by consumers' attitudes, as they decide whether to buy the newly introduced products, i.e., electric vehicles [7,15,20].

Consequently, there is an urgent need to understand consumers' perceptions and attitudes—particularly those of car drivers in Kuwait—as part of GCC and an emerging market perspective [22,23]. Therefore, it is important to examine consumers' initial viewpoints on EVs before international car companies can manufacture and export those cars for consumers in Kuwait, because car manufacturers are not present in Kuwait [22,23].

The remainder of this paper is structured as follows. We started by discussing the background of the problem and its significance, followed by the discussion and synthesis of the existing literature. We then outlined and presented our study design and the data collection process. This section is followed by a presentation and discussion of the descriptive findings, limitations of the study, managerial implications, suggestions for future research and a concluding section.

## 1.1. Problem Statement and Significance

Kuwait is a small Arab country located in the Middle East and is about one-third of the size of Scotland [24]. Kuwait has a rich economy, with crude oil reserves of nearly 6% of the world reserves. Petroleum accounts for over half of Kuwait's GDP, 92% of export revenues and 90% of its government income [25]. Kuwait is recognized as one of the most profitable car markets and the largest automobiles in the Middle East and North Africa (MENA) region, given the low cost of fuel, the tax-free customs on imported automobiles, the improved economic situation, and the increased per capita income over the last 50 years [22,23,25]. Consequently, Kuwait has the highest vehicle ownership rate in the MENA region. This proves that the conventional automobile market is growing in Kuwait and should strive to import cars that suit the needs and wants of diverse consumers [22,23].

There is an infrastructure issue in Kuwait due to the bad road conditions during severe rains in the winter season, and the high rates of accidents along with high traffic speed. During the summer season, the temperature reaches well over 50 °C degrees in Kuwait [22,23,26]. These conditions lead to high maintenance that might lead to potholes until fixed that not be suitable for EV batteries, which are generally situated at the bottom of the vehicles. Kuwait is clearly facing a multitude of challenges in curbing its greenhouse gas emissions to meet the pre-conditions of the Kyoto protocol and the COP21 targets, due to its desire for construction and reconstruction, its industrial development, its infrastructure growth, its surging population and its oil and gas activity [27]. In 2019, the greenhouse gas (GHG) emissions from Kuwait were about 136.69 (MtCO2e) or 32.49 tCO2e/person, with transport being the third highest GHG emitting sector [28]. Ground transportation only account for 12% of the total release of GHG [29]. Although that percentage might appear low in comparison to other sectors, one has to keep in mind that the emissions per capita in Kuwait ranked the second highest in the world in Kuwait after Qatar and about five times higher than the average in the European Union [28].

## 1.2. EV Adoption for GHG Reduction and Improvement of Air Quality in Kuwait

Fuel transition from fossil fuels to zero-emission vehicles, such as EVs, could significantly lower the GHG emissions. This development has already been proven in Norway, a country with similar population size to Kuwait. Norway has already replaced 20% of its ground transportation fleet for EVs and as a result achieved a 3% permanent reduction in the country's release of GHG [30]. Hence, transitions in Kuwait from conventional ICE vehicles to EV offers a tremendous opportunity to lower GHG emissions as EVs are less than 1% of total car populations [31] and offer a viable solution to Kuwait's pledge to lower GHF emissions, both to the United Nations through their Nationally Determined Contributions (NDCs) [32] and as their own national vision 2035 for the State of Kuwait,

especially sustainable goal number 13: "Take urgent action to combat climate change and its impacts" [33].

Another significance of this study would be the improvement of air quality should the transition to EV take place. Driving a zero-emission vehicle, such as an EV, eliminates harmful substances (e.g., nitrogen dioxide, carbon monoxide) from the air and drastically improves air quality in urban areas. The Natural Geographic Society [34] estimates that one-third of all air pollution is derived from ground transport. In Kuwait, however, regional pollution, sand, and dust account for over 80% of the air pollution measured in PM2.5, while pollution from traffic is estimated to contribute 16% [35]. Lowering air pollution with zero-emission vehicles does have an impact.

Consequently, the main objective of this research is to explore the EV landscape in Kuwait and provide significant insights into the phenomena and offer viable solutions on how EV adoption in Kuwait might be fruitful as a means to lower GHG emission and improve air quality [26,32].

## 2. Literature Review

The existing studies have established that consumers' attitudes towards sustainable mobility and cleaner technologies, i.e., electric vehicles, are likely to have a positive impact on environmental and social outcomes [14,36,37]. The importance of the concept of sustainable mobility is becoming the crucial point among academics, leaders, managers, and governments worldwide due to the exponentially increasing population; further, the consumption of conventionally powered fueled cars has increased substantially, accompanied with the growth of greenhouse gases (GHG) emitted by traffic [15]. Accordingly, the solution to the existing environmental problem should focus on sustainable mobility and greener and cleaner technologies, such as EVs with limited or zero GHG emissions [4]. It would also be important to pressure the market to shift towards the adoption of greener models and products by promoting sound environmental behavior and sustainable mobility through public pressure and demands from consumer interest groups and NGOs, in order to influence people to buy more eco-friendly products, such as electric vehicles. [14,36,37].

The literature about consumers' attitudes towards electric vehicles is growing [38–42]. The vehicle price, speed, purchasing cost, driving range, battery replacement, re-charging time and maintenance costs are among the important attributes and features listed in most (past and recent) studies of consumers' attitudes towards electric vehicles in America, Japan and European countries, such as Spain, UK, Germany, Norway and the Netherlands. In this regard, an extensive literature review was conducted to identify all the attributes or features of EV that have been studied around the world (i.e., developed, developing countries, and MENA region), as perceived by potential customers of EV (see Table 1).

As shown in Table 1, most studies have investigated attributes or features of EVs in developed countries, and to some degree in developing countries. However, little research has explored these attributes or features in the MENA region. Additionally, none have examined this issue via survey-based studies to provide insights into conventional car drivers' attitudes towards EVs in Kuwait. Furthermore, most studies have concentrated more on some attributes or features (i.e., price/cost, availability of recharging facilities, supportive government incentives and policies, driving range, and environmentally friendly), while some attributes or features have rarely been explored (i.e., battery life, road infrastructure, recharging time, safety and reliability, and operating condition). Moreover, some attributes or features have been explored in developed countries only (i.e., speed, technology advancement, comfort, and design). In contrast, some attributes or features have been explored in developing countries (i.e., operating condition (heater and A/C)).

**Table 1.** Attributes or features of EVs as identified by potential customers around the world.

| Attributes/Features of EV | Industrialized Countries | Developing Countries (Excluding MENA Region) | MENA Region |
|---|---|---|---|
| Price (Cost) | Thiel et al. [43], Lebeau et al. [44], Axsen and Kurani [45], Vilchez et al. [41], Ottesen and Banna [22], Peters and Dütschke [46], Zhang et al. [13], Degirmencia and Breitnerb [47], Dua and White, [48], Archsmith, Muehlegger, and Rapson [49], Mandy [50], Higueras-Castillo et al. [51,52]. | Bhalla, Ali, and Nazneen [53], Kim, Oh, Park, Joo [54,55], Khurana, Kumar, and Sidhpuria [56], Thananusak et al. [57], Dasharathraj Shetty et al. [58], De Oliveira et al. [59] | Eneizan, (Jordan) [60], Abu-Alkeir, (Jordan) [61] |
| Recharging availability/facility (public or home) | Axsen and Kurani [45], Ottesen and Banna [22], Morton, Anble, and Nelson [62], Vassileva and Campillo [63], Archsmith, Muehlegger, and Rapson [49] | Bhalla, Ali, and Nazneen [53], Haider, Zhuang and Ali [64], Kim, Choi, and Yoi, and Kim, Kim, Oh, Park, Joo [54,55], Khurana, Kumar, and Sidhpuria [56], De Oliveira et al. [59] | |
| Government Incentives/policy | Axsen and Kurani [45], Vilchez et al. [41,42] Peters and Dütschke [46], Zhang et al. [11–13], Dua and White [48], Higueras-Castillo et al. [51,52], Archsmith, Muehlegger, and Rapson [49] | Haider, Zhuang and Ali [64], Kim, Choi, and Yoi, and Kim [54,55], Khurana, Kumar, and Sidhpuria [56], Dasharathraj Shetty et al. [58], Khazaei and Tareq [65], De Oliveira et al. [59] | Al-Buenain et al. Quatar [66], Shareeda, Al-Hashimi and Hamdan, (Bahrain, GCC) [67], Kiani, (Algeria) [68] |
| Driving range | Thiel et al. [43], Lebeau et al. [44] Morton, Anble, and Nelson [62] Vassileva and Campillo [63], Degirmencia and Breitnerb [47], Higuera-Castillo et al. [51,52], Mandys [50] | Khurana, Kumar, and Sidhpuria [56], Kongklaew et al., Kongklaew et al. [69], Thananusak et al. [57] | |
| Environmentally friendly | Ziefle et al. [70], Morton, Anble, and Nelson [62], Peters and Dütschke [46], Degirmenc and Breitnerb [47], Nosi et al. [71] | Kim, Oh, Park, Joo [54,55], Khazaei and Tareq [65] | |
| Battery life/battery performance | Archsmith, Muehlegger, and Rapson [49] | Kongklaew et al. [69,72] Haider, Zhuang, and Ali [64], Kim, Choi, and Yoi, and Kim [54,55], Khazaei and Tareq [65] | |
| Road/public infrastructure | Zhang et al. [11–13], Archsmith, Muehlegger, and Rapson [49] | Kongklaew et al. [69,72] Thananusak et al. [57] De Oliveira et al. [59] | Shareeda, Al-Hashimi and Hamdan, (Bahrain, GCC) [67] |
| Recharging time | Thiel et al., Lebeau et al. [43] | Khurana, Kumar, and Sidhpuria [56], Thananusak et al. [57], De Oliveira et al. [59] | |
| Safety/trust/reliability/engine performance (i.e., low noise) | Ziefle et al. [70]), Higuera-Castillo et al., Higueras-Castillo et al. [51,52] | Kim, Choi, and Yoi, and Kim (2022) [54,55], Thananusak et al. [57], De Oliveira et al.) [59] | Abu-Alkeir, (Jordan) [61] |
| Learning/experience/awareness/knowledge | Vassileva and Campillo [63] | Kongklaew et al. (2021) [69], Dasharathraj Shetty et al. [58] | Eneizan (Jordan) [60], Shareeda, Al-Hashimi and Hamdan ((Bahrain, GCC), [67] |
| Speed | Thiel et al. [43] Lebeau et al. [44] | Thananusak et al. [57] | |

**Table 1.** *Cont.*

| Attributes/Features of EV | Industrialized Countries | Developing Countries (Excluding MENA Region) | MENA Region |
|---|---|---|---|
| Technology advancement | Zhang et al. [11–13], Dua and White [48] | Dasharathraj Shetty et al. [58] | |
| Social/subjective norms | Archsmith, Muehlegger, and Rapson [49] | | Eneizan (Jordan) [60] |
| Comfort | Ziefle et al. [70] | | |
| Design/brand | Axsen and Kurani [45] | Dasharathraj Shetty et al. [58] | Abu-Alkeir (Jordan) [61] |
| Operating condition (heater and A/C) | | Kim, Choi, and Yoi, and Kim (2022) [54,55] Thananusak et al. [57] | |

Clearly, there are insufficient studies in the MENA region. Khalifa and Maliki [73] found that new conventional fuel car-purchasing decisions in North Africa are based on brand perceptions among consumers in Algeria (MENA region), whereas another study explored the effects of cultures and traditions that shape the habits among Kuwaiti consumers, without taking into consideration EV consumers or drivers [74]. Kiani [68] examined the global EV market trends, the complementary battery technologies, and the trends in manufacturers, emission standards across borders and prioritized advancements, and found that local governmental policies in terms of technology, infrastructure requirements, changes in power dynamics, consumers' incentives, market-regulating behavior and effective communications with stakeholders are essential in the UAE. It is important to note that this study did not consider the conventional car drivers and their attitudes towards EV. Shareeda, Al-Hashimi and Hamdan [67,68] reviewed the advantages and disadvantages of electric vehicles and how they are perceived in terms of value and adoption, especially as it is a new sustainable product for the Bahrain market (GCC region). They suggested that the participation of all key stakeholders and their engagement within the innovation process is vital to develop EVs.

Al-Buenain et al. [66] compared EVs and conventional vehicles in terms of the environmental performance to evaluate the economic and practical feasibility of the use of EVs in Qatar. Their study showed that, regardless of favorable outcomes of swapping to sustainable mobility solutions, a lack of readiness predominates the Qatari automobile consumer market. Their study has highlighted the need for more sensible motivations and incentives and called for substantial grants and financial support from the government of Qatar to rise sustainable e-mobility solutions. A study conducted by Eneizan [60] used the Theory of Planned Behavior (TPB) model to show that attitudes towards the adoption of EVs, subjective norms, and perceived behavioral control positively impacted EV adoption intention among individuals in Jordan (MENA region). Similarly, Abu-Alkeir [61] examined the consumers' intentions to purchase hybrid cars in Amman-Jordan and found that price, the reputation of the manufacturer, the fuel economy, brand image and safety rating factors impacted customer intention regarding purchasing EVs. The study focused on hybrid cars and assumed electric cars are the same. It is worth noting that the weather in Jordan is much colder than in GCC countries, which makes it more suitable for the adoption of EVs. To the best of our knowledge, there was only one pilot study conducted by Ottesen and Banna [22,23] which attempted to provide an initial insight into new car purchasing behavior among consumers in Kuwait. Interestingly, the study concluded that there are three potential new car buyer segments seeking value, performance and safety. Their study failed to examine attitudes of conventional car-drivers towards EVs.

In light of the above, none of the studies have specifically explored the phenomena in Kuwait as a critical region of the Gulf countries and the MENA region. Therefore, there is a need to investigate all or some of these attributes or features in the MENA region in order to identify which of these attributes or features are considered important by

potential consumers, as this region constitutes a very promising market. Hence, the present study aimed to fill the gap in the literature by adding more significant knowledge, and contributed as follows:

1. Attitudes of conventional car drivers towards EVs in Kuwait;
2. Sustainable mobility of the EV phenomenon, particularly from the car drivers' perspectives in Kuwait;
3. Whether the attitudes of conventional car drivers in Kuwait aligned with the results yielded in industrialized countries and/or developing countries;
4. The enablers and barriers of EVs in Kuwait and, hence, provided suggestions for promoting these enablers and overcoming the barriers.

## 3. Methods

### 3.1. Research Instrument

This study utilized a mixed method approach involving the collection of quantitative data and qualitative data to achieve the study objective. A large-scale questionnaire survey was carried out to provide a broad picture of the EV phenomenon in Kuwait, while the qualitative data (only two open-ended questions) intended to cover the same ground in depth, and to confirm the survey results. A draft survey along a pilot test evaluating the links was circulated among a group of instructors at the Faculty of Business Administration at Arab Open University (AOU) to collect their feedback and incorporate their suggestions. Participants were asked to fill out the survey along with an evaluation of the survey. The pilot test evaluation asked questions about the objectives, contents, and design of the questionnaire on a Likert scale of 1 to 5 (where 1 means "strongly disagree", 3 means "neither agree nor disagree" and 5 means "strongly agree"). The draft survey along with the pilot test evaluation was distributed to 50 instructors (full-timers) and tutors (part-timers). In total, we had 30 respondents. The overall response rate was nearly 60%. Some changes and revisions have been made to the format of the questionnaire and to the overall design. For example, we have removed the sections and treat the questionnaire as one solid unit to make it easier for the respondents to fill it in. Another option, "I do not know", should be added to collect sincere answers from respondents and allow them to express their views freely. We have added one question to collect respondents' feedback and further comments about EV in Kuwait. The analysis of the pilot test evaluation shows that nearly 96% of the respondents agree that the content is clear, useful, and easy to understand. Almost 93% of the respondents agreed that the questionnaire's objectives are stated clearly, the content is sufficient to address the research question, and the sequence of the questionnaire is organized and well structured. While 90% of the respondents agree that the content is accurate and comprehensive and the design is flexible, 86% of the respondents agree that the structure of the questionnaire is easy to follow. The first part focuses on the data related to the demographic characteristics of the respondents: this covers gender, age, education, household income, employment, nationality, and numbers of cars owned by the households. The first part consists of 11 items about the demographic characteristics of the respondents. The second part of the survey consists of in-depth questions (open and closed types of questions) to achieve the purpose of the study. The second part consists of 18 closed-ended questions and 2 open-ended questions to enrich our understanding of the viewpoints of EV drivers in Kuwait.

### 3.2. Sampling Procedures and Size

Since this study aimed to collect data about EVs, the researchers had to focus on the drivers and owners of conventional cars, who are the most likely buyers of EV. Therefore, the researchers decided to use a random sampling technique to collect data. For this study, the population comprised all people who drive and/or own conventional cars in Kuwait. As the population of this study was over 100,000, the minimum required sample size was 384, as suggested by Krejcie and Morgan [75], and any sample size of over 500 was "very good", as suggested by Comrey and Lee [76]. We stopped collecting data after collecting

the questionnaires from 604 participants. The final sample included 472 individuals (78.1% response rate) who were used in the analysis.

### 3.3. Data Collection Procedure

Two weblinks were formulated: one for the Arabic version and another for the English version. The authors formulated the questionnaire using Google Forms, which is a validated tool used for data collection. The researchers started to distribute the questionnaires among several groups in Kuwait who usually drive and/or own conventional cars, such as students, the public, faculty members, and tutors, to collect their feedback and comments using two weblinks. The respondents to the questionnaire needed to be at least 18 years old and car drivers to participate in this study. The data-collection stage lasted from February 2022 until May 2022. The participants were told the purposes of the study and asked to complete the questionnaire. The instructions for completing the questionnaire were given on a cover page to avoid any misunderstanding about the issue. To ensure the objectivity of the study, the respondents were asked one qualifying question to ensure they drive and/or own a conventional car. Only after ensuring this were they allowed to answer the rest of the questionnaire. A total of 604 persons participated in this study. After removing 132 questionnaires (i.e., those who do not have or drive a car), a total of 472 questionnaires were analyzed. As Table 1 demonstrates, there was an approximate gender balance within the data (238 males and 234 females). Additionally, 64% of the sample (*n* = 304) was in the age category of 26 to 60 years, which reflects the car ownership in Kuwait.

### 3.4. Ethical Consideration

The researchers obtained ethical approval from the London School of Economics and Political Science—Middle East Center and LSE Research Ethics Committee.

### 3.5. Data Analysis

The researchers conducted statistical analysis using the SPSS software. Descriptive statistics (i.e., frequencies and percentages) were computed to analyze the data relating to the demographic characteristics and closed-ended questions. The collected data from the two open questions were analyzed using an online free-of-charge text-analyzer software named "LEXICOOL". It analyzes text and supports the word frequency by counting the words expressed by respondents. It allows researchers and end-users to analyze text by recording the counts of characters and words for each sentence by providing statistical information on the repetition of phrases and keywords. The researchers then grouped the frequencies of similar phrases altogether and provided themes plus their frequencies for the first open-ended question, and then reported the frequencies for the second open-ended question.

## 4. Findings

A summary of the demographic characteristics of the respondents is presented in Table 2. Approximately 50.4% males and 49.6% females completed the questionnaire. Approximately half of the participants (47%) belonged to the age range of to 39 years, while more than one-third of them (35.6%) belonged to the age range of 18 to 25 years. More than half of the respondents were single (57.6%), while approximately one-fifth of them (18.2%) were married with three kids or more. Approximately two-thirds (60.8%) of the participants were Kuwaiti. More than half of the respondents (54.9%) have a bachelor's degree. More than a quarter of the participants own two cars (29.4%), or five cars or more (28.6%). More than a third of the participants (37.3%) were employed in the private sector, while approximately one-third of them (32.2%) were employed in the public sector. Approximately one-third of participants (29.4%) worked in other private services, and more than a quarter of them (26.5%) worked in the government and ministries sector and more than one-fifth of them (21.6%) worked in middle management. The monthly income of more than half of the participants (53.6%) was less than KD 1000 (USD 3300).

**Table 2.** A summary of the demographic characteristics of the respondents.

| Variable | Categories | *n* = 472 | % |
|---|---|---|---|
| Gender | Male | 238 | 50.4% |
| | Female | 234 | 49.6% |
| Age Range | 18–25 years | 168 | 35.6% |
| | 26–39 years | 222 | 47.0% |
| | 40–49 years | 61 | 12.9% |
| | 50–60 years | 21 | 4.4% |
| Marital Status | Single | 272 | 57.6% |
| | Married without kids | 37 | 7.8% |
| | Married with 1 kid | 35 | 7.4% |
| | Married with 2 kids | 42 | 8.9% |
| | Married with 3 kids or more | 86 | 18.2% |
| Ethnicity | Kuwaiti | 287 | 60.8% |
| | Arab—Non-Kuwaiti | 144 | 30.5% |
| | Asian—Non-Arab | 38 | 8.1% |
| | American—European—Australian | 2 | 0.4% |
| | African—Non-Arab | 1 | 0.2% |
| Number of Cars in household | One car | 62 | 13.1% |
| | Two cars | 139 | 29.4% |
| | Three cars | 70 | 14.8% |
| | Four cars | 66 | 14.0% |
| | Five cars or more | 135 | 28.6% |
| Educational Level | Less than high school | 8 | 1.7% |
| | High school diploma | 108 | 22.9% |
| | Trade/commerce degree | 55 | 11.7% |
| | Bachelor's degree | 259 | 54.9% |
| | Master's degree | 31 | 6.6% |
| | PhD | 11 | 2.3% |
| Employment | Private sector | 176 | 37.3% |
| | Public sector | 152 | 32.2% |
| | Unemployed | 83 | 17.6% |
| | Self-employed | 35 | 7.4% |
| | Family-owned business | 26 | 5.5% |
| Field of employment | Other private services | 139 | 29.4% |
| | Government and ministries | 125 | 26.5% |
| | My family business | 61 | 12.9% |
| | Educations—government and private | 46 | 9.7% |
| | Oil and gas sector | 32 | 6.8% |
| | Large Kuwaiti corporation | 29 | 6.1% |
| | Healthcare—government and private | 26 | 5.5% |
| | Military or police | 14 | 3.0% |
| Which of the following best describes your role in industry? | Middle management | 102 | 21.6% |
| | Administrative Staff | 80 | 16.9% |
| | Upper management | 59 | 12.5% |
| | Student—not working | 54 | 11.4% |
| | Lower management | 38 | 8.1% |
| | Support Staff | 34 | 7.2% |
| | Temporary employee | 28 | 5.9% |
| | Self-employed/business partner | 27 | 5.7% |
| | Trained professional expert | 22 | 4.7% |
| | Researcher | 12 | 2.5% |
| | Consultant | 8 | 1.7% |
| | Skilled laborer | 8 | 1.7% |

**Table 2.** *Cont.*

| Variable | Categories | *n* = 472 | % |
|---|---|---|---|
| | Less than 500 | 149 | 31.6% |
| | 500–999 | 104 | 22.0% |
| Monthly Income (KD) | 1000–1499 | 111 | 23.5% |
| | 1500–1999 | 59 | 12.5% |
| | 2000 and above | 49 | 10.4% |

With regard to the attitudes of participants relating to EV, see Table 3. As shown in the table, approximately half of the participants (45.6%) would pay 6–20% more for an EV that is environmentally friendly than gasoline cars, while about a third of them (32.9%) would pay a maximum of 5% more for such a car. More than 40% of participants (43.8%) would pay 6–20% more for an EV that is much quicker than a normal gasoline car (0–100 in 4 s), while about a third of them (32.4%) would pay a maximum of 5% more for such a car. More than 40% of participants (43.5%) would seriously think about buying an EV if the gas/fuel prices increased by between 50% and 199%, while about a fifth of them (18.9%) did not care about EVs. More than half of the participants (51.5%) would change their mind towards buying an EV if the government regulated and controlled the costs of these cars at between 10% and 30% cheaper than gasoline cars, whereas about a fifth of them (19.5%) would change their mind if the cost was similar to gasoline cars. Approximately a third of participants (30.3%) would change their mind towards buying an EV if there were public and free fast-charging stations every 10 to 25 km, while approximately a quarter of them (23.3%) would change their mind if there were charging stations every 26 to 50 km. More than half of the participants (57.6%) would change their mind towards buying an EV if there was a fast lane only for EVs on major highways. More than half of the participants (54.9%) would change their mind towards buying an EV if there were public and free parking spaces at almost the same capacity as handicap spaces. More than half of the participants (53.4%) would buy an EV within the next 3 years. Of those, 14% would certainly buy one, and 39.4% were very likely to. A total of 41.7% of participants thought that EVs are safe in relation to fire and car crashes, and 39% of them would be able to charge an EV in their residential area in Kuwait, while about a third of them (32.7%) would not be able to do so.

**Table 3.** Attitudes of participants relating to EV.

| Question | Categories | *n* = 472 | % |
|---|---|---|---|
| | 0% | 83 | 17.6% |
| | 1–5% | 72 | 15.3% |
| How much more would you pay for an electric car that is environmentally friendly than gasoline cars? | 6–10% | 117 | 24.8% |
| | 11–20% | 98 | 20.8% |
| | 21–29% | 48 | 10.2% |
| | 30% and more | 54 | 11.4% |
| | 0% | 78 | 16.5% |
| | 1–5% | 75 | 15.9% |
| How much more would you pay for an electric car that is much quicker than a normal gasoline car or about 0–100 in 4 s? | 6–10% | 103 | 21.8% |
| | 11–20% | 104 | 22.0% |
| | 21–29% | 58 | 12.3% |
| | 30% and more | 54 | 11.4% |
| | 500% and above | 14 | 3.0% |
| | 400–499% | 12 | 2.5% |
| | 300–399% | 19 | 4.0% |
| How much would the gas/fuel have to increase to make you seriously start thinking about buying an electric car? | 200–299% | 50 | 10.6% |
| | 100–199% | 97 | 20.6% |
| | 50–99% | 108 | 22.9% |
| | Less than 50% | 83 | 17.6% |
| | Indifferent/I do not care about Electric cars | 89 | 18.9% |

**Table 3.** *Cont.*

| Question | Categories | *n* = 472 | % |
|---|---|---|---|
| I would change my mind towards buying and electric car if the government regulated and controlled the costs of electric cars to be__________. | 30% cheaper costs than gasoline cars | 135 | 28.6% |
| | 10% cheaper costs than gasoline cars | 108 | 22.9% |
| | The same costs as gasoline cars | 92 | 19.5% |
| | 10% higher costs than gasoline cars | 58 | 12.3% |
| | Indifferent/I do not care about Electric cars | 79 | 16.7% |
| I would change my mind towards buying and electric car if there were public and free fast-charging stations every ___________. | Less than 10 km | 85 | 18.0% |
| | 10–25 km | 143 | 30.3% |
| | 26–50 km | 110 | 23.3% |
| | 51–75 km | 41 | 8.7% |
| | 76 km and more | 23 | 4.9% |
| | Indifferent/I do not care about Electric cars | 70 | 14.8% |
| I would change my mind towards buying an electric car if there was a fast lane only for EVs on the major highways (such as on the 30 and 40 highway) | No | 96 | 20.3% |
| | Yes | 272 | 57.6% |
| | Indifferent/I do not care about Electric cars | 104 | 22.0% |
| I would change my mind towards buying an electric car if there were public and free parking spaces almost at the same capacity as handicap spaces. | No | 119 | 25.2% |
| | Yes | 259 | 54.9% |
| | Indifferent/I do not care about Electric cars | 94 | 19.9% |
| Would you buy an electric car in Kuwait within the next 3 years? | off course NOT | 71 | 15.0% |
| | Unlikely | 149 | 31.6% |
| | Very likely | 186 | 39.4% |
| | Certainly | 66 | 14.0% |
| How safe do you think electric cars are in Kuwait in terms of fire and car crash? | Very Dangerous | 30 | 6.4% |
| | Dangerous | 44 | 9.3% |
| | Neutral | 201 | 42.6% |
| | Safe | 99 | 21.0% |
| | Very safe | 98 | 20.7% |
| If you have an electric car, are you able to charge your electric car in residential areas in Kuwait? | Impossible | 81 | 17.2% |
| | Very difficult | 73 | 15.5% |
| | Neutral | 134 | 28.4% |
| | Possible | 63 | 13.3% |
| | Very possible | 121 | 25.6% |

　　　　The present study collected opinions by asking for comments and feedback about electric vehicles in Kuwait. Table 4 shows the most repeated themes, generated by text-analyzer software. Over 59% of the 472 respondents highlighted benefits relating to the environment and proposed phrases such as "green", "preserving", "eco-friendly", "air pollution", "carbon gas emission reduction", "greenhouse gases", "health improvement", and "cancer-free". Over 55% expressed their interest in the functionalities and features of EVs, such as smooth driving, powerful engine, speed, comfort, safety, and rechargeable and recyclable batteries. Nearly 50% of the respondents pointed out obstacles and expressed their worries about EVs in Kuwait in terms of speed limits, less popularity, unsuitability due to the hot summer season, the lack of government plans and policies, the lack of EV dealerships, worrying about the costs for maintenance, lacking shaded areas designed for EVs, the poor quality of roads, and the absence of a culture of environmental awareness among people in Kuwait.

　　　　A second open-ended question required respondents to express their thoughts and opinions by asking them "What is the first word that comes to your mind when you hear "Electric Vehicles in Kuwait"? Over 70% of the sample had positive attitudes towards electric vehicles (EV), expressing words such as "fun, amazing, wow, great, good, awesome, new, exciting, happy, future etc." Table 5 shows that the respondents thought of the word "Tesla" when asked about EV in Kuwait. Another repeated word was "environmentally friendly", followed by "charging", "modern technology", "money saver" and "battery".

A cloud of repetitive words was generated to produce more insight and clarification, as shown in Figure 1.

**Table 4.** Themes generated on EVs.

| | |
|---|---|
| **Benefits Relating to the Environment** | Green, preserving, eco-friendly, air pollution, carbon gas emission reduction, greenhouse gases, health improvement, cancer-free |
| **Functionalities and Features of EVs** | Driving range, powerful engine, speed, comfort, safety, rechargeable and recyclable batteries, rechargeable points/networks |
| **Obstacles and Worries** | Speed limits, less popular, not suitable for hot summer weather in Kuwait, lack of governmental plan, dealerships are more focused on fueled cars, maintenance costs, easy shaded parking, drivable roads, lacking culture of awareness of the environment |

**Table 5.** Frequent words related to EVs.

| Words | Frequency |
|---|---|
| Tesla | 56 |
| Environmentally friendly | 35 |
| Charging | 29 |
| Modern technology | 23 |
| Money saving | 10 |
| Batteries | 9 |
| Safety | 4 |

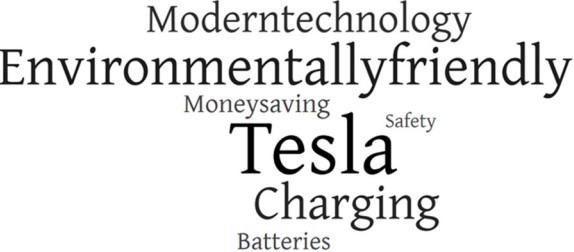

**Figure 1.** A cloud of the repeated words relating to electric cars/vehicles.

## 5. Discussion

This study aimed to examine the attitudes of drivers towards EVs in Kuwait. According to the findings, more than half of participants would buy an EV within the next 3 years, and they would buy it under four conditions: the government regulates and controls the costs of these cars to be between 10% and 30% cheaper than gasoline cars; the availability of public and free fast-charging stations every 10 km to 50 km; the availability of fast lanes only for EV on major highways; and the availability of public and free parking spaces almost at the same capacity as handicap spaces. This means that drivers in Kuwait perceived the role of the government as fundamental in encouraging the popularity of EVs, by providing suitable infrastructure and pricing policies for them. This result is consistent with previous studies in other countries [11–13,77,78].

Moreover, more than 40% of participant would seriously start thinking about buying an EV if gas/fuel prices increased by between 50% and 199%. More than 40% of participants thought that EVs are safe in relation to fires and car crashes. Furthermore, approximately half of the participants would pay 6–20% more for an EV that is both environmentally friendly and much quicker than gasoline cars. This means that drivers in Kuwait would prefer EVs over gasoline cars in the future for their environmental, economic, and technological

values. This result is consistent with those of previous studies in other countries [9,11–13]. In addition, about 40% of the respondents would be able to charge an EV in their residential area in Kuwait. This means that the infrastructure for charging EVs is available in some residential areas in Kuwait.

Additionally, we discovered the most repeated themes relating to how drivers feel and think about EVs, including environmental friendliness, the functionalities and features of EVs, and obstacles facing EVs in Kuwait. The findings also show that drivers had positive attitudes towards EVs, expressing words such as "fun, amazing, wow, great, good, awesome, new, exciting, happy, future, etc." Although most of the drivers cared about the environment and were willing to buy EV in the future, they had concerns regarding their lack of practicality due to the hot and dry weather conditions and the battery lives of EVs in Kuwait. This indicates that respondents are aware of the design of the EVs that are already available on the international market and yet are limited in an emerging market such as Kuwait.

## 6. Limitations and Future Studies

Although the present research has provided greater insights into the sustainability of the EV phenomenon by including more diverse views of consumers, such as employees, non-Kuwaitis and the self-employed, certain limitations exist. Firstly, although the present study utilized a mixed-mode approach (qualitative and quantitative questionnaire, with open-ended and closed questions), the study remained descriptive due to the lack of a focus on hypothesis testing and empirical testing. This is why a reliability test was not conducted in this study. Therefore, future studies should involve purely quantitative data and empirical investigations of various hypotheses and variables, in order to better represent the studied population in Kuwait; this will yield a more reliable conclusion about the population in Kuwait. Secondly, future studies should focus on qualitative in-depth interviews with current owners of EV in Kuwait about obstacles and opportunities, features and preferred designs, and focus group studies on what features and services would be needed for mass implementation; another comparison study about incentives to buy EVs based on experiences of other countries should be conducted [31]. Finally, future studies should be emphasized on managers of car dealerships companies in Kuwait to collect their viewpoints and obtain more insights to the electric car phenomena to further explore the obstacles for not adopting the sustainable mobility models of electric cars and being so hesitant to bring electric cars in Kuwait to replace the conventionally fueled powered cars [26].

## 7. Conclusions and Implications

The burgeoning issue of sustainable mobility has increased in recent years. Planners, leaders, and decisionmakers around the globe are faced with environmental sustainability and sustainable mobility challenges. This study provides invaluable information and insights for stakeholders in Kuwait, as an example of an emerging market. The key objective of this research is to examine and explore the sustainable mobility of the EV phenomenon in Kuwait. The results of the current study indicated that the potential customers have the intention and would buy an EV within the next 3 years; they would buy one under several conditions, including cheaper pricing policies controlled by the government and the availability of suitable infrastructure for EVs relating to charging stations, fast lanes, and free parking spaces. Potential customers in Kuwait would also prefer EVs over gasoline cars in the future for their environmental, economic, and technological value. Although there is an ultimate desire for EVs in Kuwait among diversified groups of people, there are still some major concerns, which could form barriers for EV introduction in Kuwait. One of the barriers is the lack of government plans, policy, and initiatives to build awareness and make a firmer culture focused on environmental preservation in Kuwait by introducing the sustainable mobility of EVs and discouraging the use of conventional fueled cars.

Additionally, the current study has theoretical and practical implications and suggestions. As regards theory, this study contributes to the limited literature on the sustainable mobility of EVs in the MENA region in general, and in Kuwait in particular. It could also assist researchers to compare EV adoption between developing and developed countries. As regards practice, the findings suggest that drivers in Kuwait prefer EVs over ICE cars for their environmental, economic, and technological value. Therefore, marketing campaigns should highlight these values when targeting people, particularly car drivers and owners. Moreover, this study suggests that drivers in Kuwait would buy an EV in the future if the infrastructure relating to fast-charging stations, fast lanes, and free parking spaces were to be available. Therefore, policymakers and governments are encouraged to start building and providing this infrastructure in order to encourage the adoption of EVs.

Additionally, the attractiveness of the sustainable mobility of EVs for the government and consumers in Kuwait lies in their significant environmental benefits due to their lower GHG emissions. Therefore, the adoption of EVs can reduce transport emissions and related local health risks, slow down global warming and promote the use of renewable energy. Therefore, this study suggests that governments might need to implement programs of awareness to educate customers about environmental sustainability and air pollution. Another approach is to provide subsidizing programs for EV buyers to combat the high prices of EVs. There also should be changes made to infrastructure, to include EV recharging networks throughout Kuwait. This would make it more convenient for customers to recharge their cars using renewable energy sources and not run out of electricity. Roads must be improved and cared for in order to make them more suitable for EV drivers in Kuwait [47,57].

This study discussed a variety of suggestions to overcome the barriers of EV introduction to Kuwait, and potential marketing strategies together with government initiatives and policies. The Kuwait market needs to be ready before the introduction of EVs. The old automobile industry should be prepared for transformation. Fossil fuel prices will spike, and the impact of emissions on the environment will necessitate a call for changes in individual transportation attitudes and habits. The automobile sector will drift gradually towards EVs. These cars are energy efficient, generating less greenhouse gas emissions, improving the air quality.

**Author Contributions:** A.O. is the principal investor of the project and is responsible for the conceptualization, methodology, and editing as well as funding acquisition and administration. S.B. and B.A. were responsible for the literature review, conceptualization, synthesis, methodology, validation, formal analysis, data curation, writing—original draft preparation, and editing. All authors have read and agreed to the published version of the manuscript.

**Funding:** This paper is a part of wider study called "Breaking the ICE reign: mixed method study of attitudes towards buying and using EVs in Kuwait". The study was funded by the Kuwait Foundation for the Advancement of Sciences and administrated by London School of Economics and Political Science—Middle East Center and LSE Research Ethics Committee approval.

**Institutional Review Board Statement:** This study has been approved by the London School of Economics and Political Science Ethics Committee (00558000004KJE9AAO) dated 24 November 2021.

**Informed Consent Statement:** Informed statement about usage and purpose of the study was included in the questioner as directed by LSE Ethics Committee.

**Data Availability Statement:** Not applicable.

**Conflicts of Interest:** The authors declare no conflict of interest.

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
