# Peer review of "Attitudes of Drivers towards Electric Vehicles in Kuwait"

_sustainability, doi:10.3390/su141912163_

Round 1

Reviewer 1 Report

1- Abstract: You need to briefly explain the focus or aim of your study, and the problem that raised the concern in your research.

2- What is the originality and value of your current study, especially compared with findings of previous research until now? Even though the current study explains this issue, it seems to be just explanations about what author(s) have investigated and tested without deeply evaluating prior studies related to the present topic. In this sense, the part of the Introduction, LR in this paper should be restructured to differentiate your research question and purpose from previous studies by thoroughly comparing, contrasting, and evaluating those antecedents.

3- Your problem statement requires improvement. In the current form, the discussion is patchy and less convincing. Please structure your problem one by one with facts and evidence for easy understanding.

4- What are the solid issues with evidence that attitudes of the drivers towards Electric cars in Kuwait to be investigated?  What issues triggered this situation?

5-The structure of the Literature Review is fine however the review is colloquial and lacks arguments. Please inject the elements of argument and controversy. Also, please include also more recent references to ensure that this study is current. Please put more arguments on each of the factors investigated.

6- As this study applies a quantitative study as well which distributes questionnaires. Please explain your min sample size.

7- what are the reasons select students, the public, faculty members, and tutors as your respondents?

8-in the paper mentioned various statistics. However, the author only tested simple analysis to get the findings which are frequencies and %. 

9- Please explain your data collection procedure.

Author Response

1. We used English Editing Service provided by this journal after we rewrote the paper to improve the readability as well as grammar.   2-4.  We added 2 sub chapters on the purpose and significance of this research.  5. The process and the findings were sharpened.  6 See about the significant and the research purpose.  7.  The references were all double checked, while putting it in the format that is required by this journal.  8.  Finding is clarified and in-line with the research purpose now.  9.  Data collection procedure and the limitation explained as this paper is limited in a focus as it is one of 5 paper series that all support each other and are apart of the same research grant project. 

We want to thank you for very constructive comments and to improve our paper by your review. 

Reviewer 2 Report

The following must be resolved before its publication:

1. Provide practical and research aims in the Introduction section.

2. There are no research questions. Why? They must be enumerated in the Introduction section.

3. Gaps are not outlined. To outline them more efficiently, the previous research efforts need to be provided in a table format.

4. The approach should be depicted to increase the clarity of the presentation.

5. Preferences of stakeholders from the oil industry are not taken into account. Why? The survey must cover their viewpoints. This has to be added and elaborated on.

6. References are outdated.

Author Response

1-2 As per your request two sub chapter about the significance and the purpose of this research were added.  1.1 and 1.2.    3.  We used the English Editing and Formatting Service of this journal of format the tables and summary of findings.  4. The purpose, Process, Findings and Limitation were all sharped as per your request.   5.  This study has limitation explained in the limitation chapter.  It is one paper of 5 that is under the same research project. Each paper has limited scope while all 5 have more holistic view.  As per the grant agreement all these papers will be presented to all stakeholders at the completion of this project and the feedback will be presented in a policy paper that will be presented to the London School of Economics Middle East Center as well as the Research Council of Kuwait that advices the government on GHG emission strategies and if mass implementation of EVs should be one of them.  As part of the rewriting all the references were checked and some taken out for this reason that they were outdated.  

Reviewer 3 Report

Dear Editors

Dear authors,

Thank you for the opportunity to read and review this paper. This paper describes a study that aims to examined the attitudes of drivers towards EC in Kuwait using mixed method.

Before considering publishing this paper, this paper needs several improvements.

Introduction: the authors not provide contribution of their study, novelty research and the explanation of the structure of paper in the last paragraph in introduction section.

Literature review must to separate into 2-3 sub-title (ex. Pervious studies about consumers attitudes towards EC ) to clear explanation. where do electronic cars come from? Import or domestic production? Need more explanations.

Validity and reliability test need to examine the draft of questionnaires. Because the author does not carry out validity and reliability using SPSS or AMOS, then the credibility and quality of the questionnaire is not maximal and looks subjective. this should be added to the limitations section.

Method section: author must to separate the methodology section into several parts: study instrument; data collection technique, ethical consideration, and data analysis.

When and how the authors do interview? Not explain in method section

Besides, The references not fit with journal template

Author Response

We thank you for your comments and review.  These points are constructive and have improved our journal entry.  We took all the reviewers comments under considerations and tried to respond to each and everyone of them and rewrote our paper accordingly.  After this rewriting we used the English Editing and Formatting Service provided by this journal to improve grammar and formats as well as the literature review as instructed.  This review also included setup the references again according to the requirement of this journal (9). While going changing the references we used the change to cross check them and reformat as well as delete some of them that we considered outdates.   (1) As requested we added two sub chapters on the significance and purpose of the paper, which is look at a viability to introduce a program of Mass Implementation of EVs as a mean to reduce the country's GHG emissions.  However, this paper is a part of a larger study that is managed by LSE Middle East Center and Back funded by Kuwaiti Research Council that advices the government on GHG reduction strategies.  As stated in the limitation part this paper is only a part of series of several papers that form a wholistic view of the EV adaptions possibilities in Kuwait.  Each paper is limited in scope but as a whole they form holistic view that will be presented to the Research Council as an input to policy and will be presented to all relevant stakeholders that will result in a policy paper.   The methodology of the date sampling is better explained as well as the validity and the data processing as per using SPSS etc.  Your comments have greatly improved this article and we are very satisfied after the revisions and I hope you will be too as we think all your concerns have been met. 

Reviewer 4 Report

The paper entitled „Attitudes of Drivers towards Electric Cars in Kuwait” is devoted to study the attitudes of drivers towards Electric Cars in Kuwait. While I found this paper interesting reading, there are still a few issues which the authors should take into account during the paper review.

Not all your claims are supported by the references; for instance, if one considers the sentence ”The burgeoning issue of sustainable mobility has received an increased attention in recent years”, the lack of the relevant references is obvious issue. Therefore, please revise your paper carefully and provide the relevant references.

Findings: this is not a place to describe the sample; by default, the sample is described in the Methodology Section, or in your case, in the Method Section. By nature, the Findings Section aims to answer the formulated research question. However, in the experimental research the authors describe the collected data, along with the data analysis methods.

please provide the coding schema;

please describe the “text analyzer software” which you used in your study, and provide justification of using it;

please generated the cloud of words since such image will provide the general notion of the research goal (see the title of your paper); there are a few on-line and free-of-charge tools which you can use.

please format your paper accordingly to the journal guidelines;

In general, the paper is well-written and well-organized. 

Author Response

We do appreciate your review that has greatly improved our article.   As requested we added 2 subsections about the significance and the purpose of this article which is one of many (as explained in the limitations) to support the claim that we believe the mass implementation of EVs in Kuwait will significantly lower its GHG emission in Kuwait.  This paper is part of the series to form a practical implementation guide to the National Research Council that funds these research.   For the formatting issues you mentioned we used the English and Formatting Service that is provided by this journal after we rewrote the paper according to your comments and others.  During this formatting we double checked the references and took out some outdated ones and added other to strengthen  our claims as requested.  The process of the data selection and analysis is improved per our request, as well as the findings and their implications in support of the aim of this paper and the research in general. We are very proud of this article now and we hope you like it too.  

Round 2

Reviewer 2 Report

The paper is significantly improved. I have no additional comments.

Reviewer 3 Report

Can be published in current form

Reviewer 4 Report

Dear Authors,

thank you for addressing all the issues raised.

In my opinion, the current version of your manuscript can be considered for the publication.